# Influence of Electron Beam Irradiation on the Moisture and Properties of Freshly Harvested and Sun-Dried Rice

**DOI:** 10.3390/foods9091139

**Published:** 2020-08-19

**Authors:** Lihong Pan, Jiali Xing, Xiaohu Luo, Yanan Li, Dongling Sun, Yuheng Zhai, Kai Yang, Zhengxing Chen

**Affiliations:** 1Key Laboratory of Carbohydrate Chemistry and Biotechnology, Ministry of Education, Jiangnan, Wuxi 214122, China; yelanplh@outlook.com (L.P.); lyn19881012ph@163.com (Y.L.); 6190112167@stu.jiangnan.edu.cn (Y.Z.); yangkai164@outlook.com (K.Y.); zxchen@jiangnan.edu.cn (Z.C.); 2National Engineering Laboratory for Cereal Fermentation Technology, Jiangnan University, Wuxi 214122, China; 3Jiangsu Provincial Research Center for Bioactive Product Processing Technology, Jiangnan University, Wuxi 214122, China; 4Collaborative Innovation Center for Food Safety and Quality Control in Jiangsu Province, Jiangnan University, Wuxi 214122, China; 5Ningbo Institute for Food Control, Ningbo 315048, China; hellojiali77@gmail.com; 6Wuxi EL PONT Radiation Technology Co., Ltd., Wuxi 214151, China; sundl_1980@163.com

**Keywords:** Electron Beam Irradiation, rice, moisture, physicochemical properties

## Abstract

Moisture content is an important factor that affects rice storage. Rice with high moisture (HM) content has superior taste but is difficult to store. In this study, low-dose electron beam irradiation (EBI) was used to study water distribution in newly harvested HM (15.03%) rice and dried rice (11.97%) via low-field nuclear magnetic resonance (LF-NMR). The gelatinization, texture and rheological properties of rice and the thermal and digestion properties of rice starch were determined. Results showed that low-dose EBI could change water distribution in rice mainly by affecting free water under low-moisture (LM) conditions and free water and bound water under HM conditions. HM rice showed smooth changes in gelatinization and rheological properties and softened textural properties. The swelling power and solubility index indicated that irradiation promoted the depolymerization of starch chains. Overall, low-dose EBI had little effect on the properties of rice. HM rice showed superior quality and taste, whereas LM rice exhibited superior nutritional quality. This work attempted to optimize the outcome of the EBI treatment of rice for storage purposes by analyzing its effects. It demonstrated that low-dose EBI was more effective and environmentally friendly than other techniques.

## 1. Introduction

Rice (*Oryza sativa* L.), the primary staple for more than half the world’s population, is produced worldwide. Approximately 90% of rice is grown in Asia. From 2000 to 2013, the rice productivity of the U.S. increased by an estimated 29% or approximately 2.2% on an annual basis; this increase is second only to the increase in peanut productivity (at 3.5% annually) among major U.S. field crops [1]. China, as the world’s largest rice producer and consumer, produced 199 million tons and consumed 203 million tons of rice between October 2018 and September 2019. Rice stocks reached 176 million tons in 2018. Rice consumption accounts for the largest grain ration, 81.96%, in China [2].

Considering the increasing levels of rice stocks, storing rice safely and securely has become a concern of researchers because improper storage easily results in considerable social and economic losses due to aging and insect infestation [3,4,5]. A good method for grain storage is necessary to maintain good rice quality in the face of massive rice stocks. Farmers generally use sunlight drying to reduce the moisture content of rice to extend storage time. However, although moisture reduction extends rice storage time, it will also reduce rice taste by increasing hardness [6,7]. Previous studies have shown that cooked rice grains have the best textural quality when they are stored with a moisture content of 15.70% [5]. People are increasingly seeking high-quality experiences along with improvement in life quality. Therefore, how to extend the storage time of rice while retaining its high-quality taste has become a thorny problem. Existing rice storage methods include natural drying, cryogenic storage, controlled-atmosphere storage, and chemical storage [8,9,10]. However, these methods have their drawbacks. Physical methods, such as natural drying and cryogenic storage, all require low rice moisture content. Although these methods can prolong rice storage time, achieving good taste quality is difficult. Controlled-atmosphere storage technology is a green environmental protection technology that is unsuitable for high-moisture (HM) grain because it causes alcohol accumulation and affects rice quality. Chemical methods can cause the accumulation of harmful substances and harm human health.

For several decades, food irradiation technology has been widely used to preserve food, such as fruit and meat [11,12]. In traditional food irradiation processing, the energy of gamma radiation produced by a radioisotope (CS-137 or Co-60) is transferred to irradiated food, killing the eggs and microorganisms that the food contains and thus extending shelf life [13]. Radionuclides have the disadvantage of continuous gamma ray emission, which can be a potential source of environmental hazards for operation staff and for some equipment. Electron beam irradiation (EBI), another food irradiation technique, is produced by an electron accelerator. In contrast to gamma irradiation, it can be switched off during off-duty hours without causing harm to personnel and equipment. The United Nations Food and Agriculture Organization suggested that the appropriate use of radiation for food decontamination is safe and that the irradiation of any food commodity with an overall average dose of up to 10 kGy presents no toxicological hazard and no special nutritional or microbiological problems [14].

Given that the EBI treatment of rice has no moisture requirement, newly harvested and non-dried rice can be irradiated. According to previous studies [5], an excellent texture and edible quality of rice are maintained by using this treatment. Our research team found that low-dose EBI has little effect on the quality of rice [15] while simultaneously extending storage time. In this study, EBI was applied to treat freshly harvested paddy rice and sun-dried rice, and the cooking quality and moisture migration of irradiated rice and the thermodynamics and digestion properties of the isolated starches were investigated. This work attempted to optimize the outcome of the EBI treatment of rice for storage purposes by analyzing the effects of this treatment method. It demonstrated that low-dose EBI was more effective and environmentally friendly than other methods.

## 2. Materials and Methods

### 2.1. Materials

Directly harvested paddy rice (15.03% moisture content) and sun-dried paddy rice (11.97% moisture content) were supplied by a peasant household (Nantong, Jiangsu, China). Pancreatic α-amylase (EC 3.2.1.1, 10 units/mg) and amyl glucosidase (EC 3.2.1.3, 3260 units/mL) were purchased from Megazyme (Wicklow, Ireland).

### 2.2. Preparation of EBI Paddy Rice

A total of 500 g of paddy rice was placed in a polyethylene bag and spread to a thickness of 1 cm. Paddy rice was irradiated at doses of 0, 1, 2, 3, or 4 kGy in separate batches by using an industrial electron accelerator operated at a dose rate of 2 kGy/s. The energy of accelerated electrons was 10 Mev, and the beam current was 1.0 mA with 1000 mm scan widths. After irradiation, all samples were stored in a desiccator at room temperature (~25 °C) for further analysis.

### 2.3. Low-Field Nuclear Magnetic Resonance Measurement

A MesoMR23-060V-I NMR spectrometer was used to determine moisture migration in irradiated rice (MesoMR23-060V-I, Niumag Co., LLC., Suzhou, China). To avoid the evaporation of water, 2 g of rice grains were weighed and rapidly placed into sample bottles that were then sealed and incubated at 25 °C for 30 min. At the same time, other rice grains were soaked in deionized water for 30 min. Surface moisture was wiped off from the rice grains, which were rapidly placed into sample bottles that were then sealed and incubated at 25 °C for 30 min to observe moisture migration. The Carr–Purcell–Meiboom–Gill pulse sequence scanning experiment was performed with the following parameters: number of sampling points TD = 14,992, number of repeated scans NS = 8, TW (ms) = 2000, NECH = 500, relaxation decay time DR = 3 s, with a 90° pulse of 7.52 ms, and 90°–180° pulse spacing of 14.48 ms. The inverse Laplace transformation of T2 curves was performed by using the associated software of the instrument.

### 2.4. Pasting Property of Rice

The pasting properties of rice flour were determined by using a Rapid Visco-Analyzer (RVA 4500, Perten Co. Ltd., Sydney, Australia). Rice paste (3.0 g dry basis, 25.0 g of deionized water) was held at 50 °C for 1 min and heated to 95 °C at a heating rate of 6 °C/min. The paste was held at 95 °C for 5 min and then cooled back to 50 °C at the same rate. Finally, the paste was held at 50 °C for 2 min. The peak, hold, and final viscosity values were obtained from viscograms.

### 2.5. Analysis of Rheological Properties

After the analysis of pasting properties, the corresponding dynamic and static rheological properties of each rice sample were determined as follows:

A rheometer (DHR-3 rheometer, Waters Co. Ltd., Milford, MA, USA) was used to analyze the viscoelastic properties of each rice sample. In brief, starch paste (approximately 4 mL) was loaded onto the lower plate. The upper parallel plate (40 mm diameter) was slowly lowered to reduce the gap between the two parallel plates to 1.0 mm prior to the run.

Dynamic viscoelasticity measurement: The test temperature was set at 25 °C, the scan strain was set to 1%, and the frequency was set from 0.1 Hz to 100 Hz to obtain the sample elastic modulus (G′), loss modulus (G″), and loss tangent (Tan δ = G″/G′). The changes in viscoelasticity during the sol-gel transition were indicated by the G′ value.

Dynamic shear force recovery test: The temperature was set at 25 °C constantly, and the shearing rate ranged from 0.1 s^−1^ to 100 s^−1^. The shear structure was recorded as the apparent viscosity.

### 2.6. Swelling Power and Solubiliy Index

Rice samples (0.5 g) were transferred into pre-weighed centrifuge tubes with 20 mL of distilled water. The rice flour suspensions were then incubated in a water bath for 30 min at 60 °C, 70 °C, 80 °C, and 90 °C with vortexing after every 5 min. After cooling the samples to room temperature, the tubes were centrifuged at 5000× *g* for 15 min. The supernatant was decanted in a preweighed aluminum specimen box. The weight gain of the centrifuge tubes was expressed as the swelling index. Moisture dishes were dried at 105 °C for 12 h and then cooled in a desiccator to room temperature. The weight gain of the moisture dishes was expressed as the solubility index.

### 2.7. Instrumental Texture Profile Analysis of Cooked Rice

The textural properties of the cooked rice were measured in accordance with the method of Li et al. [16] with some modifications. A total of 30 g of rice and 42 g of water were weighed and stewed in a rice cooker for 30 min. After the rice was cooled to room temperature, an 8.0 g subsample of cooked rice grains was weighed and placed in a single layer on the base plate and tested by using a TA.XTPlus Texture Analyzer (SMS Co. Ltd., Godalming, UK). A cylindrical probe (P/35) was applied with the following test speeds: pretest speed = 1 mm/s, test speed = 1 mm/s, and post-test speed = 1 mm/s. The strain was 60%, and the rice samples were compressed twice. Textural parameters, including hardness, cohesiveness, springiness, resilience, and chewiness, were calculated from the curves that were provided by the equipment.

### 2.8. Sensory Evaluation of Cooked Rice

The sensory analyses of cooked rice were performed by five trained panelists. The following sensory attributes were evaluated: taste, appearance, softness, flavor, and cold rice texture. The sensory evaluation of rice flours was conducted in accordance with the Chinese National Standard GB/T 15682-2008 [17]. Briefly, 10 g of rice was weighed out and rinsed twice with distilled water. Water was added to the sample at 1.3 times the sample volume. The sample was soaked at 25 °C for 30 min. Then, it was cooked in a steamer for 40 min and simmered for 20 min. The different samples were placed on a white porcelain plate. The reviewer tasted the sample while it was hot. Scoring was as follows: 25 points for taste, 20 points for flavor, 20 points for appearance, 30 points for softness, and 5 points for cold rice texture.

### 2.9. Isolation of Rice Starch

Rice starch was isolated via traditional alkaline extraction. Rice grains were ground and soaked in deionized water for 2 h and subsequently milled for 2 min by using a colloid mill. The mixture was subsequently centrifuged to remove water and then soaked in 0.2% sodium hydroxide solution for 48 h. The fluid was changed every 12 h. The filtrate was repassed through a 100-mesh sieve (pore size of 0.15 mm) and centrifuged at 6000× *g* for 10 min. Then, the upper gray matter (protein) was scraped off (repeated five times). The pH was adjusted to 7.0 by using dilute hydrochloric acid. This process was repeated five times to remove salt from the starch by using deionized water through centrifugation at 6000× *g* for 10 min. This process was followed by freeze–drying for 48 h. The obtained starch samples were stored in a desiccator for further analysis.

### 2.10. Differential Scanning Calorimetry

The thermal properties of starch samples were measured by using a differential scanning calorimeter (DSC3, Mettler Toledo Co. Ltd., Greifensee, Switzerland). Sample pretreatment was performed in reference to the protocol used by Zhou et al. [18] with some modifications. In a 40 µL capacity aluminum pan, distilled water was added to starch (3.0 mg) to obtain a starch suspension containing 70% water. The hermetically sealed pans were maintained at 4 °C for 12 h to ensure the equilibration of the starch and water before differential scanning calorimetry (DSC) analysis. The samples were scanned from 30 °C to 100 °C at a heating rate of 10 °C/min with an empty pan used as reference.

### 2.11. In Vitro Digestion

The digestibility of the starch samples was mainly determined in accordance with the method of Englyst and Wang [19,20] with some modifications. A 100 mg starch sample was added into a screw-cap tube and treated with 4 mL of digestive juice. The remaining steps were consistent with those of Wang’s method.

### 2.12. Statistical Analysis

All data were processed by using an Origin 9.0 and presented as mean ± standard deviation (SD). Data were statistically analyzed via one-way analysis of variance and Duncan’s multiple range test by using SPSS statistics 17.0. Differences were considered statistically significant at the 95% level (*p* < 0.05). Each sample was measured in triplicate.

## 3. Results and Discussion

### 3.1. Analysis of Water Status via Low-Field Nuclear Magnetic Resonance

Rice samples with low moisture (LM) content (11.97%) and high moisture (HM) content (15.03%) were subjected to low-field nuclear magnetic resonance (LF-NMR). The proton distributions in the control and irradiated rice samples were compared, and the results are presented in Figure 1. The control and irradiated rice samples were soaked simultaneously for 30 min for comparison, and the results are presented in Figure 2. The relative peak areas of the less- and more-mobile water fractions in each sample are displayed in Appendix A. The T_2_ relaxation time in the curves reflects the chemical environment encountered by protons in the sample, and the relaxation times of T_21_, T_22_, and T_23_ can be considered to be related to bound water, nonflowing water, and free water in paddy rice, respectively [21].

The intensity in T_21_ (0.01–1 ms) was significantly different between freshly harvested rice and sun-dried rice. After irradiation, the bound water content of the two types of rice increased at the same time, and the free water content decreased. However, the proportion of the reduction shown by the free water content of LM rice was higher than that shown by the free water content of HM rice. In contrast to our study, a previous study showed that EBI treatment reduces the bound water content of egg white protein [22]. This discrepancy might be attributed to the different proteins present in eggs and rice. As shown in Figure 1a, the bound water content of freshly harvested rice was higher than that of sun-dried rice. During sun drying, part of the bound water was likely to have been converted into free water that evaporated under the influence of high temperature, leading to a reduction in the proportion of the bound water. After irradiation, the free water proportion of LM rice decreased more than that of the rice with HM content. LM rice mainly contained a large amount of free water, whereas HM rice contained a small amount of free water and a large amount of bound water. Irradiation mainly affected free water in LM rice but affected free water and bound water in HM rice. Therefore, free water in LM rice was greatly influenced by irradiation and mainly turned into free radicals. However, the rice with HM content contained more bound water than the rice with LM content. As free water was decomposed by irradiation, the electron beam also cleaved the double helixes of starches and proteins. This effect generated puncture pores on microcosmic surfaces and caused changes in the quantities of bound water and free water by changing their states through reducing hydrogen bonding such that the structure increased in flexibility and decreased in compactness. These effects thus changed water distribution.

As shown in Figure 2 and Appendix A, after 30 min of soaking, almost all the LM and HM rice showed a shortened T_21_ and prolonged T_22_. The change in relaxation time (T_2_) in LF-NMR can reflect the degree of freedom of water in the sample and the chemical environment encountered by the protons of the sample. A short T_2_ (0.01–1 ms) is indicative of the tight binding degree between water molecules and matter and a low degree of freedom of water. It can be considered to be related to chemically bound water. A long T_2_ is associated with weak binding force and a high degree of freedom, and it can be considered to be related to nonflowing water (1–10 ms) and free water (10–100 ms) [23]. Nonflowing water represents physically adsorbed water, i.e., water that is trapped or present in cellular structures and does not easily flow. Free water refers to water that can flow freely and that exhibits the lowest binding force and strongest mobility. Rice is generally soaked before cooking mainly to improve the speed and effect of gelatinization. This study aimed to explore whether irradiation and different water content systems could affect the water absorption efficiency of rice and thus ultimately affect the cooking quality of rice. As shown in Figure 2, irradiation after soaking in the LM state did not affect the value of T_21_. By contrast, after soaking in the HM state, the T_21_ of irradiated rice moved to the right. This change indicated that the effect of irradiation tended to weaken the binding strength of organic matter and water. This result might be related to the effects of irradiation on free water and bound water in HM rice as mentioned above. In general, compared with that in rice that had not been exposed to the sun, the combination of water and protein starch in rice that had been exposed to the sun had lower strength, harder tissue, tighter networks, and lower water absorption rate. At the same time, irradiation had a positive effect on the water absorption rate of rice. Therefore, rice with a high water content and that had been treated with a specific dose of irradiation had the best water absorption rate and superior cooking quality.

### 3.2. Pasting Properties of Rice Samples

The RVA profiles of the rice flour samples are shown in Figure 3, and the parameters, including the peak viscosity (PV), holding viscosity (HV), final viscosity (FV), and pasting temperature (PT) of rice flour gelatinization are depicted in Appendix A. Irradiation treatment resulted in a significant (*p* < 0.05) reduction in the PV of all the samples. Sultan et al. also reported a reduction in the HV of rice upon irradiation treatment [24]. Figure 3 shows that the PV, HV, and FV of LM rice were lower than those of HM rice. This result was consistent with the research results of Ali et al., who found that starches extruded under HM (22%) conditions have higher viscosity than those extruded under LM (14%) conditions [25]. The irradiation effect in the HM state is weaker than that in the LM state. Simultaneously, no significant difference in the PV of HM rice under 1 and 2 kGy (*p* < 0.05) irradiation was observed because the husks protected the rice grains from high irradiation doses. Electron beams destroy the interior structure of rice [26] and are likely to first act on water molecules [27]. Their action on a large number of water molecules results in the production of free radicals and a certain amount of thermal energy. Moreover, free radicals and energy cause the chemical bonds of biological macromolecules to break by acting on biological macromolecules. HM rice had more water molecules than LM rice. Most of the electron beam energy acted on water molecules, thus causing limited damage to starch macromolecules and small changes in viscosity.

### 3.3. Rheological Properties

The rheological properties of natural starches and irradiated starches are depicted in Figure 4. The magnitudes of the elastic component, G′, and viscous component, G″ of the rice starch pastes increased with scanning frequency, and G′ was significantly higher than G″, which led to the absence of crossover between the two moduli over the range of applied frequencies, indicating that these parameters were all associated with weak gel properties [28]. This study revealed that both moduli of the gel prepared with irradiated rice samples were lower than those of the gel prepared with normal rice samples, whereas LM rice showed more reductions than HM rice. EBI treatment might have destroyed the continuous network structures and resulted in the changes in the internal structure of the rice kernel, thus weakening the gel network of the rice paste. These results were consistent with previously reported results [29]. The formation of the gel network in irradiated rice samples could be related to RVA results. Yu and Wang [30] stated that radiation can generate free radicals in starch macromolecules. These free radicals are capable of hydrolyzing chemical bonds and breaking large molecules into small dextrin fragments, destroying the gel network structure and consequently decreasing the apparent viscosity of irradiated rice flour. The apparent viscosity of HM rice flour changed more smoothly than that of LM rice flour likely because, given the large number of water molecules, irradiation energy first acted on water molecules instead of biomacromolecules.

### 3.4. Swelling Power and Solubility Index

The swelling power and solubility index were assessed over the temperature range of 60 °C–90 °C and are presented in Table 1. The swelling power of all the rice flours increased significantly (*p* < 0.05) over the temperature range of 60 °C–90 °C. The values for LM rice ranged from 3.20 g/g to 14.37 g/g and those for HM rice ranged from 3.58 g/g to 13.40 g/g. The swelling power was observed to be a function of temperature, with the highest value observed at 90 °C [31]. In this study, the effect of irradiation dose on swelling power was not significant and only slightly increased this parameter. In addition, the general swelling index of rice with HM content was slightly higher than that of rice with LM content. This difference was not significant. A previous study showed that the swelling power decreases because irradiation causes the depolymerization of amylopectin chains in starch molecules [32]. The swelling power slightly increases under a low irradiation dose. The slight opening of amylopectin chains under low-dose irradiation may have promoted water molecule entry, which leads to the increase in swelling power.

The solubility index of all the rice flours increased slightly over the temperature range of 60 °C–90 °C. However, with respect to that of native flour, the solubility index of irradiated flours significantly increased (*p* < 0.05) over the range of 60 °C–90 °C. The solubility index of flours is largely attributed to the presence of soluble molecules, such as amylose, sugars, and albumins, in the flour [33]. The increased solubility of irradiated samples could be ascribed to the increased depolymerization of starch chains that may result in the enhanced hydration of particles [34].

### 3.5. Textural Properties of Cooked Rice

Texture profile analysis is one of the most important tests used to examine food texture. Table 2 shows the hardness, adhesiveness, springiness, cohesiveness, gumminess, chewiness, and resilience of cooked rice. The hardness of the rice irradiated at the dose of 2 kGy was higher than that of cooked nonirradiated rice and gradually increased from 4155.1 to 4266.5. The hardness of cooked rice is mainly affected by the molecular size of amylose and the ratio of amylose branches with DPs of 1000–2000. Moreover, the stickiness of cooked rice is negatively correlated with amylose content. Thus, the increased hardness of cooked irradiated rice (2 kGy) observed in this study might be attributed to the slight reduction in the molecular size of amylose. In general, rice with low water content was hard, whereas rice with high water content was soft. Irradiation reduced textural indexes, and texture was gradually destroyed with the increase in irradiation dose.

### 3.6. Sensory Properties of Cooked Rice

The sensory properties of cooked native and irradiated rice are presented in Figure 5. As seen in the figure below, HM rice had better sensory ratings than LM rice. Compared with that of nonirradiated rice, the aroma of irradiated HM rice decreased with the increase in irradiation dose and that of irradiated LM rice had almost disappeared. Compared with the control group, the HM rice irradiated at 1 and 2 kGy did not show significant changes in taste and continued to present light fragrance and sweet taste. LM rice did not have the same taste as HM rice. Unsatisfactory rice taste might be ascribed to the production of undesirable flavors as a result of the following: irradiation produces free radicals, which break rice starch molecules [27], expose hydrophobic side chains inside proteins [35], and partially oxidize protein and fat. In general, the quality and taste of HM rice were superior to those of LM rice.

### 3.7. Thermal PROPERTIES

The DSC results for native and irradiated rice starches are presented in Appendix A and Figure 6, respectively. The figure shows that irradiation had a great influence on the thermodynamic properties of rice under LM condition but had little effect on HM rice. The heat absorption peak shifted to the left, and T_O_, T_P_, and T_C_ decreased in the LM state. Evan’s theory [36] states that the degradation of the amorphous region of starch granules decreases the stability of the crystalline region and leads to the reduction in gelatinization temperature. Irradiation caused the degradation of starch molecules, thus reducing stability and resulting in the decrement in gelatinization temperature. Cooke et al. [37] pointed out that DSC results are more likely to show the change in crystal structure than the double helix structure of starch, indicating that irradiation has destroyed part of the double helix structure of starch.

### 3.8. In Vitro Digestion

The nutritional properties of the starches were evaluated via in vitro digestion. The rapidly digestible starches (RDS), slowly digestible starches (SDS), and resistant starches (RS) levels of native and irradiated rice are presented in Table 3. The results indicated that the digestive changes shown by all irradiated samples were not significant. In general, the RDS of rice samples with HM content were significantly higher than those of rice samples with LM content, whereas the RS of rice samples with HM content were significantly lower than those of rice samples with LM content. After consumption, RDS induces hyperglycemic response and insulin resistance, which easily cause several diet-related chronic diseases and metabolic syndromes; by contrast, RS is fermented only by microorganisms in the large intestine to produce short-chain fatty acids, which are beneficial to intestinal health [19,38]. Therefore, according to the results of this study, rice with LM content has better nutritional quality than rice with HM content.

## 4. Conclusions

The results of this study suggested that low-dose EBI treatment changed water distribution in newly harvested rice with HM (15.03%) and dried rice (11.97%). Radiation mainly affected the free water components of LM rice and the combined water and free water components of HM rice, as well as resulting in the destruction of biomolecules in both types of rice. EBI treatment reduced the viscosity of rice, and the change in LM rice was more obvious than that in HM rice. Rheological moduli also changed after irradiation, and the change in HM rice was gradual. The increase in swelling power after irradiation might be attributed to the slight opening of amylopectin chains under low-dose irradiation. This effect promoted the entry of water molecules. The increased solubility of irradiated samples could be ascribed to the increased depolymerization of starch chains that resulted in enhanced particle hydration. Low-dose irradiation had little effect on the gelatinization, texture, and rheological properties of most rice samples. The quality and taste of HM rice were superior, whereas the nutritional quality of LM rice was superior. This work was an attempt to optimize the outcome of the EBI treatment of rice for storage purposes by analyzing its effects. It showed that low-dose EBI was more effective and environmentally friendly than other methods.

## Figures and Tables

**Figure 1 foods-09-01139-f001:**
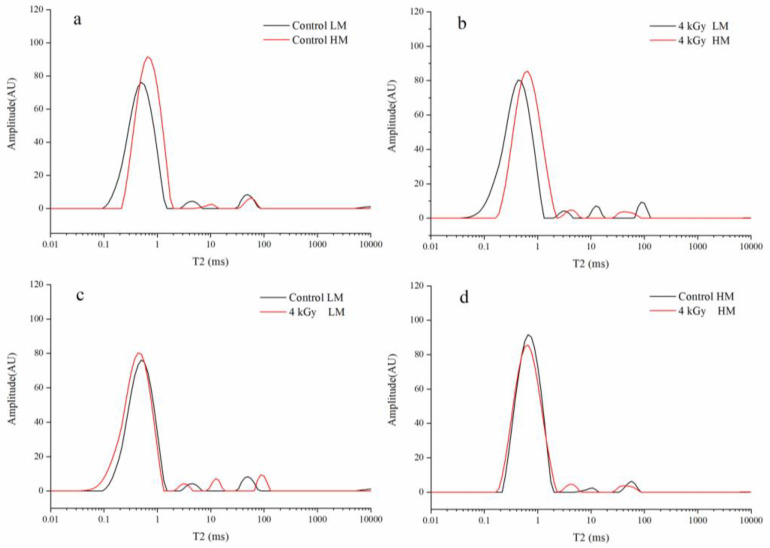
Water distribution in low moisture (LM) and high moisture (HM) rice samples under different irradiation doses. (**a**) Control LM and HM rice, (**b**) 4 kGy LM and HM rice, (**c**) Control and 4 kGy LM rice, (**d**) Control and 4 kGy HM rice.

**Figure 2 foods-09-01139-f002:**
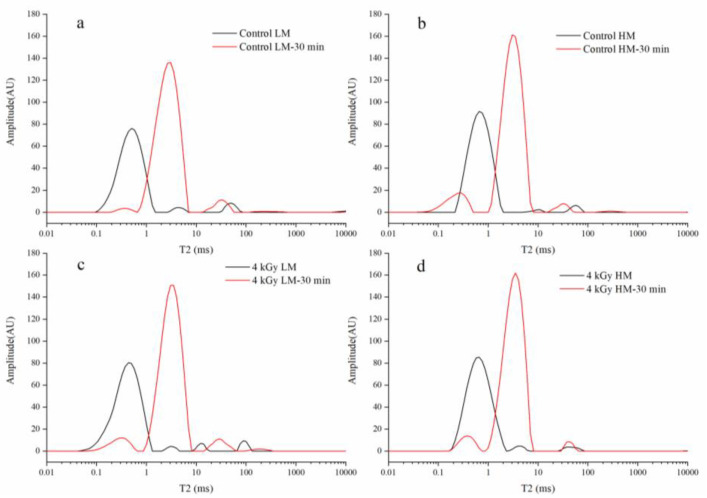
Water distribution in LM rice and HM rice samples after 30 min of soaking and treatment with different irradiation doses. (**a**) Control LM rice before and after 30 min soaking, (**b**) Control HM rice before and after 30 min soaking, (**c**) 4 kGy LM rice before and after 30 min soaking, (**d**) 4 kGy HM rice before and after 30 min soaking.

**Figure 3 foods-09-01139-f003:**
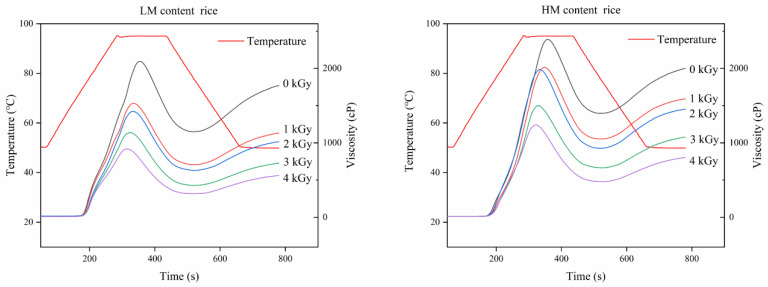
Rapid Visco-Analyzer (RVA) pasting profiles of LM and HM rice samples under different irradiation doses.

**Figure 4 foods-09-01139-f004:**
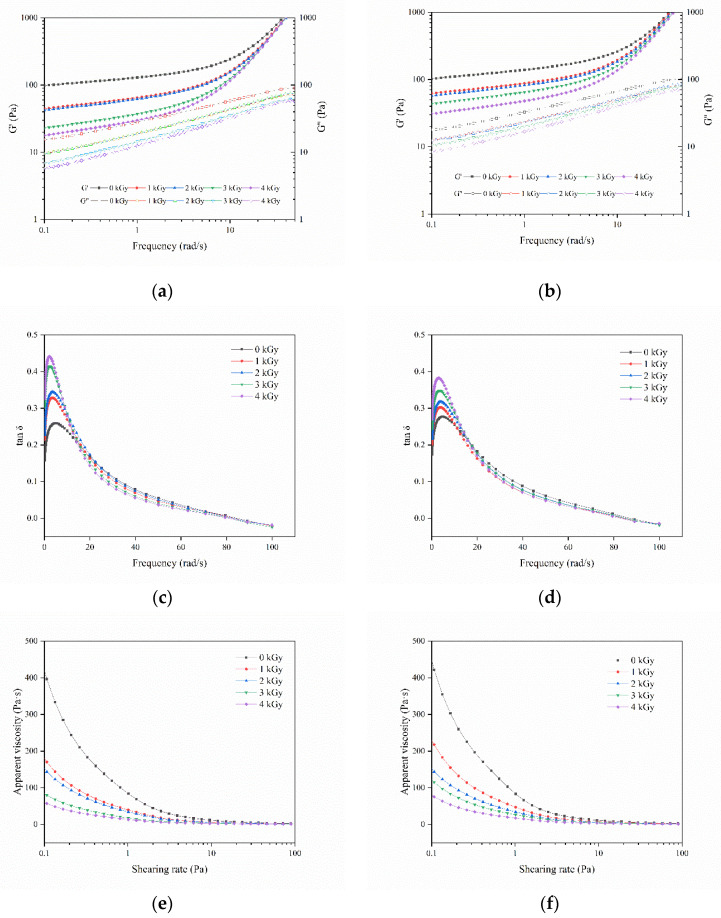
Changes in G′, G″ (**a**,**b**), tan δ (**c**,**d**), and apparent viscosity (**e**,**f**) of RSs under different electron beam irradiation (EBI) doses. (**a**,**c**,**e**) LM rice; (**b**,**d**,**f**) HM rice.

**Figure 5 foods-09-01139-f005:**
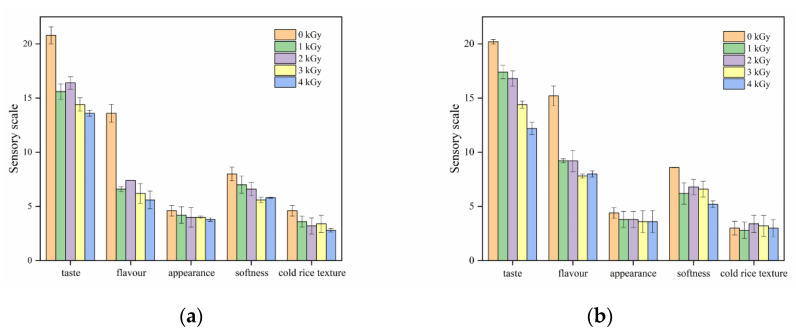
Sensory evaluation of cooked rice treated with various irradiation doses. (**a**) LM rice, (**b**) HM rice.

**Figure 6 foods-09-01139-f006:**
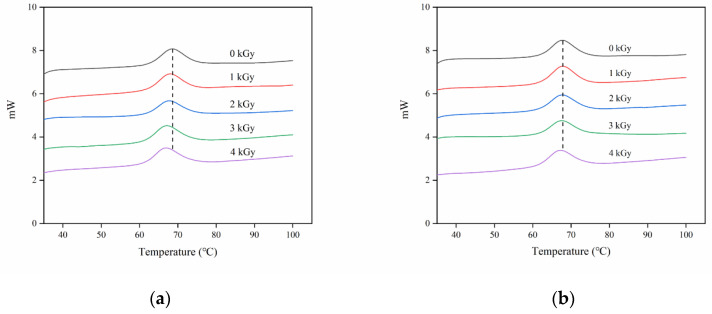
Differential scanning calorimeter (DSC) thermograms of the starches isolated from rice grains treated with various irradiation doses. (**a**) LM rice, (**b**) HM rice.

**Table 1 foods-09-01139-t001:** Swelling power and solubility index of control and irradiated rice flours *.

Varieties	Dose	60 °C	70 °C	80 °C	90 °C
SP (g/g)					
LM content	0	3.20 ± 0.04 ^ap^	7.24 ± 0.06 ^aq^	8.40 ± 0.01 ^ar^	12.07 ± 0.04 ^as^
	1	3.40 ± 0.02 ^cp^	7.63 ± 0.12 ^bq^	8.76 ± 0.10 ^br^	12.69 ± 0.41 ^abs^
	2	3.44 ± 0.02 ^cp^	7.71 ± 0.04 ^bcq^	8.85 ± 0.09 ^br^	13.90 ± 0.37 ^abs^
	3	3.28 ± 0.02 ^bp^	7.78 ± 0.04 ^bcq^	9.01 ± 0.12 ^bcr^	14.10 ± 0.71 ^abs^
	4	3.54 ± 0.01 ^dp^	7.94 ± 0.06 ^cq^	9.20 ± 0.00 ^cr^	14.37 ± 0.87 ^bs^
HM rice	0	3.58 ± 0.03 ^ap^	7.28 ± 0.08 ^aq^	8.20 ± 0.10 ^ar^	11.33 ± 0.17 ^as^
	1	4.12 ± 0.09 ^bp^	7.24 ± 0.06 ^aq^	8.33 ± 0.02 ^ar^	11.94 ± 0.25 ^abs^
	2	4.05 ± 0.11 ^bp^	7.32 ± 0.08 ^aq^	8.52 ± 0.32 ^ar^	12.43 ± 0.64 ^abcs^
	3	4.12 ± 0.04 ^bp^	7.25 ± 0.01 ^aq^	8.51 ± 0.08 ^ar^	12.75 ± 0.13 ^bcs^
	4	3.74 ± 0.19 ^abp^	7.79 ± 0.23 ^bq^	8.65 ± 0.09 ^ar^	13.40 ± 0.02 ^cs^
SI (g/100 g)					
LM content	0	5.70 ± 0.06 ^ap^	9.21 ± 0.14 ^aq^	10.11 ± 0.13 ^as^	10.44 ± 0.01 ^ar^
	1	6.74 ± 0.34 ^bp^	15.54 ± 0.03 ^bq^	15.55 ± 0.32 ^bq^	21.83 ± 0.85 ^br^
	2	6.89 ± 0.32 ^bcp^	16.45 ± 0.02 ^cq^	16.79 ± 0.25 ^bq^	23.32 ± 0.73 ^br^
	3	7.79 ± 0.30 ^cp^	19.56 ± 0.30 ^dq^	19.77 ± 0.57 ^cq^	29.48 ± 1.44 ^cr^
	4	9.47 ± 0.14 ^dp^	21.04 ± 0.11 ^eq^	22.53 ± 0.25 ^dr^	32.84 ± 1.34 ^cs^
HM rice	0	5.11 ± 0.19 ^ap^	7.40 ± 0.24 ^aq^	8.81 ± 0.20 ^as^	8.07 ± 0.06 ^ar^
	1	6.35 ± 0.55 ^bcp^	12.35 ± 0.09 ^bq^	12.56 ± 0.42 ^bq^	16.95 ± 0.26 ^br^
	2	6.14 ± 0.10 ^bp^	13.77 ± 0.07 ^cq^	13.03 ± 0.78 ^bq^	19.95 ± 0.91 ^cr^
	3	7.31 ± 0.19 ^cdp^	15.71 ± 0.20 ^dq^	15.75 ± 0.37 ^cq^	24.49 ± 0.28 ^dr^
	4	7.83 ± 0.08 ^dp^	18.15 ± 0.12 ^eq^	18.93 ± 0.11 ^dr^	29.89 ± 0.19 ^es^

* Mean value ± SD with different superscript letters in the same column are significantly different (*p* < 0.05). SP = swelling power: SI = solubility index. The first letter represents vertical significance and the second letter represents horizontal significance.

**Table 2 foods-09-01139-t002:** Textural properties of cooked rice after EBI treatment at different doses *.

Dose/kGy		Hardness	Adhesiveness	Springiness	Cohesiveness	Gumminess	Chewiness	Resilience
LM rice	0	4155.1 ± 523.1 ^a^	−1046.9 ± 148.4 ^a^	0.72 ± 0.07 ^a^	0.26 ± 0.01 ^a^	1111.6 ± 168.6 ^a^	833.6 ± 57.3 ^ab^	0.08 ± 0.00 ^a^
	1	3254.8 ± 486.7 ^b^	−672.6 ± 96.6 ^b^	0.60 ± 0.01 ^b^	0.27 ± 0.03 ^a^	899.5 ± 226.4 ^ab^	537.2 ± 142.7 ^b^	0.08 ± 0.01 ^a^
	2	4266.5 ± 494.1 ^a^	−710.3 ± 20.0 ^b^	0.63 ± 0.03 ^ab^	0.27 ± 0.02 ^a^	1172.9 ± 186.8 ^a^	731.5 ± 83.3 ^a^	0.09 ± 0.01 ^a^
	3	2793.0 ± 460.3 ^b^	−441.1 ± 32.6^c^	0.61 ± 0.04 ^b^	0.25 ± 0.02 ^a^	699.8 ± 118.5 ^b^	422.7 ± 54.7 ^b^	0.07 ± 0.01 ^a^
	4	2520.5 ± 219.1 ^b^	−477.7 ± 54.2 ^c^	0.60 ± 0.00 ^b^	0.26 ± 0.02 ^a^	649.7 ± 73.2 ^b^	395.6 ± 95.1 ^b^	0.07 ± 0.00 ^a^
HM rice	0	2476.0 ± 274.8 ^a^	−602.8 ± 82.9 ^a^	0.64 ± 0.02 ^a^	0.25 ± 0.01 ^ab^	608.7 ± 55.4 ^a^	387.9 ± 35.1 ^a^	0.06 ± 0.00 ^a^
	1	2355.8 ± 81.3 ^a^	−581.4 ± 36.1 ^a^	0.58 ± 0.06 ^ab^	0.25 ± 0.00 ^ab^	592.0 ± 32.0 ^a^	346.5 ± 47.9 ^ab^	0.07 ± 0.00 ^a^
	2	2074.3 ± 170.1 ^ab^	−375.6 ± 40.8 ^b^	0.53 ± 0.03 ^bc^	0.25 ± 0.00 ^ab^	517.1 ± 59.0 ^ab^	273.3 ± 45.7 ^bc^	0.08 ± 0.00 ^b^
	3	1692.7 ± 147.4 ^bc^	−327.0 ± 10.9 ^b^	0.47 ± 0.02 ^bc^	0.26 ± 0.01 ^a^	438.5 ± 15.6 ^b^	207.3 ± 14.7 ^cd^	0.08 ± 0.00 ^b^
	4	1427.8 ± 109.4 ^c^	−311.3 ± 25.3 ^b^	0.51 ± 0.05 ^c^	0.24 ± 0.00 ^b^	338.1 ± 18.1 ^c^	173.7 ± 26.7 ^d^	0.07 ± 0.00 ^ab^

* Mean value ± SD with different superscript letters in the same column are significantly different (*p* < 0.05).

**Table 3 foods-09-01139-t003:** In vitro digestibility of the starches isolated from rice grains under various irradiation doses *.

Dose (kGy)		In Vitro Digestibility ^#^		
		RDS (%)	SDS (%)	RS (%)
LM rice	0	28.21 ± 0.61 ^a^	47.50 ± 0.73 ^a^	24.29 ± 0.12 ^ab^
	1	30.01 ± 0.49 ^ab^	46.30 ± 0.73 ^a^	23.69 ± 0.24 ^ab^
	2	32.08 ± 1.22 ^ab^	47.41 ± 0.85 ^a^	20.51 ± 2.07 ^a^
	3	28.38 ± 3.29 ^a^	40.27 ± 0.73 ^b^	31.35 ± 4.02 ^c^
	4	33.97 ± 0.73 ^b^	39.50 ± 2.56 ^b^	26.53 ± 1.83 ^bc^
HM rice	0	34.75 ± 0.85 ^a^	47.93 ± 0.85 ^a^	17.32 ± 0.00 ^bc^
	1	35.44 ± 0.61 ^a^	48.53 ± 0.49 ^a^	16.03 ± 0.12 ^b^
	2	36.77 ± 0.12 ^a^	49.48 ± 0.37 ^a^	14.05 ± 0.49 ^a^
	3	37.33 ± 2.80 ^a^	41.99 ± 3.41 ^b^	20.68 ± 0.61 ^c^
	4	36.98 ± 0.37 ^a^	45.18 ± 1.58 ^ab^	17.84 ± 1.22 ^d^

* Mean value ± SD with different superscript letters in the same column are significantly different (*p* < 0.05). ^#^ RDS, SDS, and RS represent rapidly digestible, slowly digestible, and resistant starches, respectively.

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
