# Peer review of "Influence of Electron Beam Irradiation on the Moisture and Properties of Freshly Harvested and Sun-Dried Rice"

_foods, 2020, doi:10.3390/foods9091139_

Round 1
Reviewer 1 Report
In the paper "Influence of electron beam irradiation on the 2 moisture and properties of freshly harvested and 3 sun-dried rice", the authors discuss important aspects of storing rice with higher water content. They showed that the use of EBI treatment changed the water distribution in rice, while not changing other properties of the product. Research is well planned, conclusions are drawn correctly.
I suggest only using the word "higher" than "greater" in scientific papers.
Author Response
Response : Thank you for your reminder. The word “greater” has been replaced with “higher” (rows 192, 259, and 302).

Reviewer 2 Report
To this reviewer, the current version of the manuscript has some language issues in describing findings, discussing results, and drawing conclusions, but these could be considered relatively minor as they did not blur the presentation of the ideas. On the technical side, however, these exist a number of inconsistencies and deficiencies that can not be overlooked.
1) The last sentence of the Introduction section is problematic. It stated that "the results of this study may provide an effective and environmentally way to store freshly harvested rice with high quality." But EBI appears to be a proven method already and endorsed by UNFAO. This work actually represents an attempt to optimize the outcome of using EBI to treat rice for storage purpose by analyzing its effects.
2) The authors separated the LF-NMR relaxation times into three parts as a means to identify three different states of water in rice and to compare their amount changes. It would be better if the authors move the brief explanation between manuscript Lines 195 and 198: "The change in relaxation time....of freedom [22]" to be combined with the sentence starting Line 175, and further provide/define the ranges of the relaxation times for nonflowing and free water.
3) Line 179 stated that after irradiation, the bound water (T21) content of both types of rice increased. However, this appears to be supported by the LM case presented in Fig. 1(c), but NOT by the HM case in Fig. 1(d).
4) Between Line 187 and Line 193, the authors tried to explain their LF-NMR results but unfortunately made some incorrect or unclear statements. Firstly, accepting that LM and HM rice both have free water and that irradiation acts mostly on free water first, it still does not explain why there is a bigger portion of free water in LM rice being reduced. Secondly, bound water and free water should be different states of water, not different regions inside rice structure. So one does not flow into the other one. In addition, breaking hydrogen bonds is not possible as they continues to exist among water molecules regardless of them in either bound or free state, and between water-biomacromolecules. But it is possible that the irradiation energy can be passed onto water and biomacromolecules to cause them to be less bound by hydrogen bonds so that the structure becomes more flexible and less compact.
5) The discussion about Fig. 2 between Line 194 and Line 201 is very unclear. It also failed to explain what is the purpose of soaking in this study.
6) Each of Fig. 3 has two different vertical axes for two different types of curves. They should be better labeled. Arrows pointing to the corresponding axes or using distinguishable colors
7) The authors stated between Lines 218 and 220 that EBI "destroy" rice interior structure and HM rice suffers less damage to biomolecules. These require either direct evidences, which were not included, or better fundamental verification, which was not provided. If EBL acts first and mostly on free water, it seems counter-intuitive to have rice biomolecular structure being destroyed. Perhaps some internal heat conduction, or even heat convection, can take place for water to pass on energy to biomolecules for their structures to be "altered" or "opened up".
8) In Line 287, ".. irradiation energy acts on the 'crystalline and amorphous' regions of starch granules". To this reviewer and perhaps most readers as well, crystalline and amorphous together represent every region in the structure. It is not clear what the authors really meant to say here. Similarly, in Line 275, "... low water content is 'harder and softer' than...". How can something be harder and softer simultaneously than the other?
Author Response
To this reviewer, the current version of the manuscript has some language issues in describing findings, discussing results, and drawing conclusions, but these could be considered relatively minor as they did not blur the presentation of the ideas. On the technical side, however, these exist a number of inconsistencies and deficiencies that can not be overlooked.
Point 1: The last sentence of the Introduction section is problematic. It stated that "the results of this study may provide an effective and environmentally way to store freshly harvested rice with high quality." But EBI appears to be a proven method already and endorsed by UNFAO. This work actually represents an attempt to optimize the outcome of using EBI to treat rice for storage purpose by analyzing its effects.
Response 1: Thank you for your advice. After considering your suggestion, we made some modifications to the last sentences of the Abstract, Introduction, and Conclusions. We highlighted the edited sentences in red (lines 31–33, 78–81, and 372–375). We hope that you will find the revisions satisfactory.
Point 2: The authors separated the LF-NMR relaxation times into three parts as a means to identify three different states of water in rice and to compare their amount changes. It would be better if the authors move the brief explanation between manuscript Lines 195 and 198: "The change in relaxation time....of freedom [22]" to be combined with the sentence starting Line 175, and further provide/define the ranges of the relaxation times for nonflowing and free water.
Response 2: Thank you for your valuable comment. We have discussed and explained this part in detail. We combined line 175 and modified lines 195–198. Our revisions have been highlighted in red as follows: The change in relaxation time (T2) in LF-NMR can reflect the degree of freedom of water in the sample and the chemical environment encountered by the protons of the sample. A short T2 (0.01–1 ms) is indicative of the tight binding degree between water molecules and matter and a small degree of freedom of water. It can be considered to be related to chemically bound water. A long T2 is associated with weak binding force and a high degree of freedom, which can be considered to be related to nonflowing water (1–10 ms) and free water (10–100 ms) [23]. Nonflowing water represents physically adsorbed water, that is, water that is trapped or present in cellular structures and does not flow easily. Free water refers to water that can flow freely and that exhibits the lowest binding force and strongest mobility (see Rows 213 to 221).
Point 3: Line 179 stated that after irradiation, the bound water (T21) content of both types of rice increased. However, this appears to be supported by the LM case presented in Fig. 1(c), but NOT by the HM case in Fig. 1(d).
Response 3: Thank you very much for your careful review. Although the figure appears to show that the bound water content of HM rice has decreased, the proportion of the peak area in the figure represents the proportion of water in this state. In fact, according to Table S1 in our supplementary materials, the bound water content of HM rice has increased.
Table S1. Differences in proton areas and peak times among rice samples with different moisture contents.
Sample |
Control LM rice |
Control LM rice-30 min |
4 kGy LM rice |
4 kGy LM rice-30 min |
Control HM rice |
Control HM rice-30 min |
4 kGy HM rice |
4 kGy HM rice-30 min |
Peak 1 |
739.12±3.47b |
66.93±48.307a |
850.37±17.72c |
97.19±1.43a |
817.94±11.47bc |
97.60±48.25a |
842.57±0.99c |
81.52±3.49a |
Peak 2 |
18.47±0.28a |
1236.67±17.85b |
21.62±6.44a |
1268.45±4.73c |
14.14±5.27a |
1310.41±13.42d |
19.88±0.88a |
1309.32±2.36d |
Peak 3 |
37.49±2.46cd |
69.79±1.29e |
45.54±5.56d |
72.25±2.90e |
22.50±0.75ab |
39.47±3.77cd |
20.65±1.06d |
30.25±0.36bc |
T1 |
1.44±0.01b |
0.51±0.10a |
1.29±0.00b |
0.59±0.01a |
1.82±0.00c |
0.47±0.00a |
2.01±0.14c |
0.60±0.02a |
T2 |
5.34±0.94ab |
6.88±0.58b |
3.23±0.60a |
7.18±0.06b |
7.30±2.24b |
7.18±0.06b |
4.90±0.16ab |
7.18±0.06b |
Point 4: Between Line 187 and Line 193, the authors tried to explain their LF-NMR results but unfortunately made some incorrect or unclear statements. Firstly, accepting that LM and HM rice both have free water and that irradiation acts mostly on free water first, it still does not explain why there is a bigger portion of free water in LM rice being reduced. Secondly, bound water and free water should be different states of water, not different regions inside rice structure. So one does not flow into the other one. In addition, breaking hydrogen bonds is not possible as they continues to exist among water molecules regardless of them in either bound or free state, and between water-biomacromolecules. But it is possible that the irradiation energy can be passed onto water and biomacromolecules to cause them to be less bound by hydrogen bonds so that the structure becomes more flexible and less compact.
Response 4: Thank you for your valuable comment! We have carefully read these sentences and made some modifications. First, following your suggestion, we added some content (rows 202 to 206). Second, in accordance with your advice, we made some modifications to the last sentence (rows 207 to 211). We hope that our revisions are satisfactory.
Point 5: The discussion about Fig. 2 between Line 194 and Line 201 is very unclear. It also failed to explain what is the purpose of soaking in this study.
Response 5: We are very grateful for your helpful comment. We have considered the discussion on Fig. 2 in depth. We have edited this section and highlighted our edits in red (lines 213 to 234). We hope that our edits are satisfactory.
Point 6: Each of Fig. 3 has two different vertical axes for two different types of curves. They should be better labeled. Arrows pointing to the corresponding axes or using distinguishable colors
Response 6: Thank you very much for your careful review. We have modified Fig. 3 and added a legend to distinguish the two different vertical axes (line 258). We hope that you will find the revisions satisfactory.
Point 7: The authors stated between Lines 218 and 220 that EBI "destroy" rice interior structure and HM rice suffers less damage to biomolecules. These require either direct evidences, which were not included, or better fundamental verification, which was not provided. If EBL acts first and mostly on free water, it seems counter-intuitive to have rice biomolecular structure being destroyed. Perhaps some internal heat conduction, or even heat convection, can take place for water to pass on energy to biomolecules for their structures to be "altered" or "opened up".
Response 7: Thank you for your valuable comment! We have reviewed the literature and added the reference “Physicochemical, structural, and functional properties of native and irradiated starch: a review.” This reference provided direct evidence for the mechanisms by which irradiation destroys water molecules and biomolecules. The following figure illustrates the mechanism of irradiation. We have also modified the original text (rows 251 to 256). We hope that you will find our revisions satisfactory.
Point 8: In Line 287, ".. irradiation energy acts on the 'crystalline and amorphous' regions of starch granules". To this reviewer and perhaps most readers as well, crystalline and amorphous together represent every region in the structure. It is not clear what the authors really meant to say here. Similarly, in Line 275, "... low water content is 'harder and softer' than...". How can something be harder and softer simultaneously than the other?
Response 8: Thank you very much for your careful review. We have revised these sentences (line 336–338, 310–311). Once again, thank you very much for your comments and suggestions.

Reviewer 3 Report
Presented manuscript is interesting and indicate how electron beam irradiation influences on the properties of freshly harvested and 3 sun-dried rice. However, I have few remarks which should be taken into consideration.
Lines 75-76: Authors stated that the cooking quality of rice was performed. Which test were used to cooking quality evaluation of rice?
How long was rice stored after irradiation and before testing? How long the irradiation effect last? It is a constant effect or it can change during rice storage?
In conclusions and in summary authors stated that that quality and taste or HM rice were superior. However, authors did not study in this paper the taste of rice. If quality of rice was studied, the sensory evaluation of rice should be included.
Author Response
Presented manuscript is interesting and indicate how electron beam irradiation influences on the properties of freshly harvested and 3 sun-dried rice. However, I have few remarks which should be taken into consideration.
Point 1: Lines 75-76: Authors stated that the cooking quality of rice was performed. Which test were used to cooking quality evaluation of rice?
Response 1: Thank you for your valuable comments! We performed texture profile analysis to determine rice cooking quality. The main results and discussion are provided in Section 3.5. Table 2 shows the hardness, adhesiveness, springiness, cohesiveness, gumminess, chewiness, and resilience of cooked rice.
Point 2: How long was rice stored after irradiation and before testing? How long the irradiation effect last? It is a constant effect or it can change during rice storage?
Response 2: We are very grateful for your valuable comments. Our study was conducted shortly after irradiation. Our team is still conducting research and experiments on the effect of irradiation duration and changes during storage.
Point 3: In conclusions and in summary authors stated that that quality and taste or HM rice were superior. However, authors did not study in this paper the taste of rice. If quality of rice was studied, the sensory evaluation of rice should be included.
Response 3: Thank you very much for your advice. In accordance with your suggestion, we have added a section on sensory evaluation section to our paper. The main contents are in Sections 2.8 and 3.6 (rows 143–152, 316–329).

Round 2
Reviewer 3 Report
OK